# Quantitative Inversion Ability Analysis of Oil Film Thickness Using Bright Temperature Difference Based on Thermal Infrared Remote Sensing: A Ground-Based Simulation Experiment of Marine Oil Spill

Meiqi Wang [1], Junfang Yang [1,2,*], Shanwei Liu [1], Jie Zhang [1,2,3], Yi Ma [2,3] and Jianhua Wan [1]

1   College of Oceanography and Space Informatics, China University of Petroleum, Qingdao 266580, China
2   Technology Innovation Center for Ocean Telemetry, Ministry of Natural Resources, Qingdao 266061, China
3   First Institute of Oceanography, Ministry of Natural Resources, Qingdao 266061, China
*   Correspondence: yangjunfang@upc.edu.cn; Tel.: +86-1786-422-9460

**Abstract:** Oil spills on the sea surface have caused serious harm to the marine ecological environment and coastal environment. Oil film thickness (OFT) is an important parameter for estimating oil spills amount, and accurate quantification of OFT is of great significance for rapid response and risk assessment of oil spills. In recent years, thermal infrared remote sensing has been gradually applied to quantify the OFT. In this paper, the outdoor oil spill simulation experiments were designed, and the bright temperature (BT) data of different OFTs were obtained for 24 consecutive hours in summer and autumn. On the basis of the correlation analysis of OFT and bright temperature difference (BTD) between oil and water, the traditional regression fitting model, classical machine learning model, ensemble learning model, and deep learning model were applied to the inversion of OFT. At the same time, inversion results of the four models were compared and analyzed. In addition, the best OFT inversion time using thermal infrared was studied based on 24-h thermal infrared data. Additionally, the inversion results were compared with the measured results; the optimal OFT range detectable using thermal infrared was explored. The experimental results show that: (1) Compared with ensemble learning model, traditional regression fitting model, and classical machine learning model, Convolutional Neural Network (CNN) has the advantages of high stability while maintaining high-precision inversion, and can be used as the preferred model for oil film thickness inversion; (2) The optimal time for OFT detection is around 10:00 to 13:00 of the day, and is not affected by seasonal changes; (3) During the day, thermal infrared has good detection ability for OFT greater than 0.4 mm, and weak detection ability for thinner oil films; (4) At night, thermal infrared has certain detection ability for relatively thick oil film, but the accuracy is lower than that in the daytime.

**Keywords:** thermal infrared remote sensing; oil film thickness inversion; bright temperature difference; machine learning; oil spill amount estimation

## 1. Introduction

The frequent occurrence of oil spill accidents at sea has caused serious harm to the marine environment and marine living resources [1–4]. Timely and accurate determination of the amount of oil spilt provides a basis for rapid response to oil spill accidents [5,6]. The estimation of the amount of oil spilt includes determining the oil spill range, oil film thickness (OFT) and oil density, among which the oil spill range and oil density can be well determined in existing studies, but the quantitative inversion of OFT has always been a hot spot and frontier problem that domestic and foreign researchers pay attention to and solve [7–9]. Grasping accurate crude oil film distribution and thickness information is of great significance for emergency response and risk assessment of oil spills [10].

The advantages of remote sensing technology in oil spill monitoring are obvious [11–16]. Synthetic aperture radar (SAR) is the most commonly used sensor in oil spill monitoring, which has the advantages of all-day and all-weather [17–20], but it is easily affected by oil film analogues and is difficult to use for OFT detection [13,21,22]. Ultraviolet (UV) remote sensing is sensitive to extremely thin oil films and can detect OFT of less than 0.05 μm, but it is greatly affected by environmental factors and cannot be detected at long distances [23–25]. Hyperspectral remote sensing has high spectral resolution and can provide accurate spectral information of ground features [26,27], and has been widely used in recent years for quantitative analysis of OFT [28,29], but it is vulnerable to solar flares [5,30–32].

In addition to the above methods, due to the different heat capacity and thermal infrared emissivity of oil film and seawater, there is a brightness temperature difference (BTD) between oil film and background seawater, the detection of oil spills on the sea surface during the day and night can be realized by thermal infrared remote sensing [10], and the different OFT presents different light and dark contrast on the thermal infrared images, and the detection of different OFT can be realized according to this difference [3,33–35]. Thermal infrared remote sensing is not affected by solar flares. However, it cannot identify nonemulsified oil and oil–water emulsion, and is susceptible to interference with targets that have similar thermal properties to crude oil. Lu et al. obtained the brightness temperature (BT) data of the oil film for 26 consecutive hours by designing ground experiments, and further simulated the BTD between oil and water by using the daily temperature cycle model; the experimental results showed that the optimal detection time of OFT was around noon [33]. However, due to the uneven diffusion of the oil film in the experiment, it was impossible to obtain relatively accurate OFT data for the verification of quantitative inversion accuracy.

In this paper, through the design of an outdoor oil spill simulation experiment, the BT data of oil film with different thickness for 24 consecutive hours in summer and autumn are obtained, and the correlation analysis between OFT and BTD is carried out. On this basis, the traditional regression fitting model, the classical machine learning model, and the ensemble learning model are applied to the inversion of OFT, and then the inversion results were evaluated using the accuracy evaluation index. We would like to answer the following questions:

- In terms of OFT inversion, does the deep learning model have an advantage over the traditional regression fitting model, classical machine learning model, and ensemble learning model? An optimal OFT inversion model is determined through comparative analysis.
- What is the optimal time for OFT detection using thermal infrared in a day? Does it change with the seasons?
- For the 17 OFTs set in the experiment, how is the detection ability of thermal infrared remote sensing?
- At night, how is the OFT detection ability of thermal infrared?

## 2. Data and Methods

### 2.1. Data Acquisition and Processing

Two outdoor OFT detection experiments were conducted in summer and autumn; the experiment period was from 12:00 on 2 August 2022 to 11:00 on 3 August 2022 (24 consecutive moments in summer) and from 16:00 on 5 September 2022 to 15:00 on 6 September 2022 (24 consecutive moments in autumn). The experiments were conducted on sunny days and wind conditions (1.2–14.7 km/h) less than level 4. The experimental oil product was Dongying Shengli Oilfield crude oil (component analysis: wax 7.56, gum 18.77, asphaltene 1.57 wt%; freezing point: −9 °C). Due to the weak volatility of crude oil, the change of OFT and emulsification reaction during the observation period were ignored [33]. Experimental setup (Figure 1): 18 black circles with an inner diameter of 7 cm were used to form different OFTs; Testo 890-2 portable thermal infrared imager (spectral range: 8–14 μm,

sensitivity NETD $\leq$ 0.04 K, photo resolution 640 $\times$ 480) was used to obtain the BT images of water without and with oil film coverage; and an anemometer (Instrument name: Kestrel 5000; Measurement accuracy: 0.1 km/h) was used to record the meteorological conditions during the experiment.

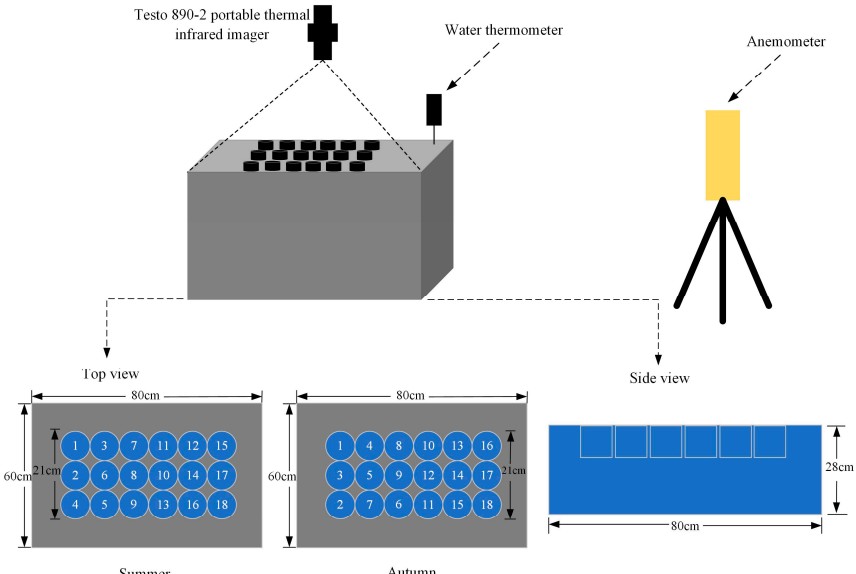

**Figure 1.** Schematic diagram of oil film thickness detection in small scenes.

In the experiment, different masses of crude oil were dripped into the rings to form different OFTs (Table 1). The rings float on the surface of the water (natural seawater) to ensure adequate heat exchange between oil and water. Crude oil density was measured in advance under laboratory conditions. We waited for the oil film to spread evenly over the entire circle, set up the portable thermal imager next to the tank, adjusted the angle of the lens so that the entire tank was within the field of view of the lens, and acquired data every one hour. In the summer experiment, a total of 84 effective thermal infrared images were acquired, and in the autumn experiment, a total of 75 effective thermal infrared images were obtained. The experimental site diagram was shown in Figure 2.

**Table 1.** Oil film thickness setting.

| Number | OFT Setting (Summer)/mm | OFT Setting (Autumn)/mm | Number | OFT Setting (Summer)/mm | OFT Setting (Autumn)/mm |
|---|---|---|---|---|---|
| 1 | 0.00 | 0.00 | 10 | 0.61 | 0.60 |
| 2 | 0.01 | 0.01 | 11 | 0.70 | 0.70 |
| 3 | 0.04 | 0.04 | 12 | 0.80 | 0.80 |
| 4 | 0.07 | 0.07 | 13 | 0.90 | 0.90 |
| 5 | 0.10 | 0.10 | 14 | 1.01 | 1.01 |
| 6 | 0.20 | 0.20 | 15 | 1.50 | 1.50 |
| 7 | 0.30 | 0.30 | 16 | 2.00 | 2.00 |
| 8 | 0.40 | 0.40 | 17 | 2.51 | 2.50 |
| 9 | 0.50 | 0.50 | 18 | 3.00 | 3.04 |

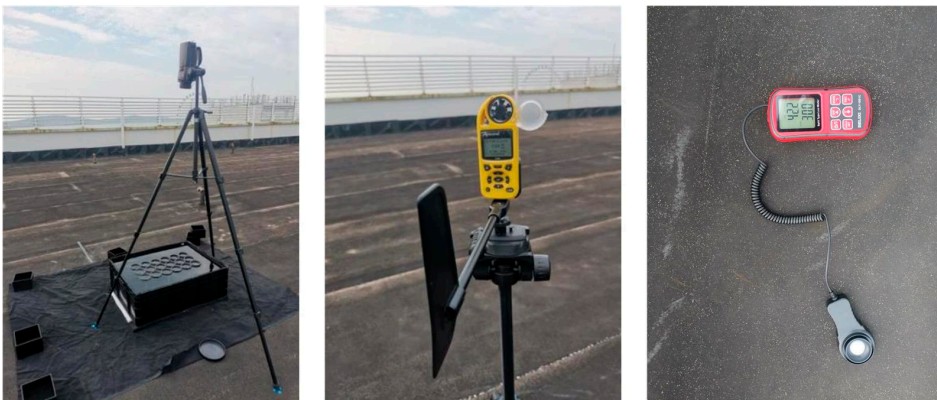

**Figure 2.** Experimental site diagram.

In this experiment, the BT data of thermal infrared covering the water surface with and without oil film were obtained, and it was necessary to correct it by using meteorological data to obtain the thermal infrared BT images shown in Figure 3. Due to the influence of the solar altitude angle and azimuth angle during the experiment, there would be shadows inside some circles, the shadows would have a great impact on the BT value on the thermal infrared images, and it was necessary to use ArcGIS software to select the available experimental areas. The samples selected were shown by the colored squares in Figure 4B. The BT values in the selected experimental area for each circle were averaged to represent the BT values at that thickness. We obtained the BT value corresponding to each pixel in the selected experimental region and its corresponding actual OFT value, and made the thermal infrared BT and BTD data sets of different OFT.

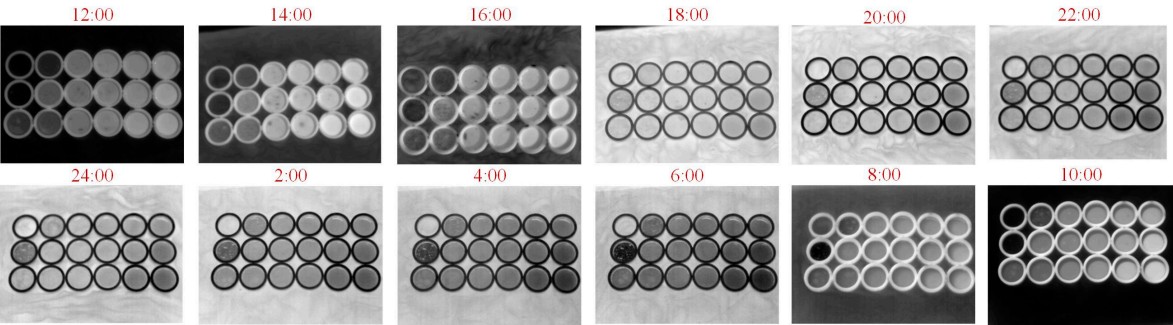

**Figure 3.** Thermal infrared bright temperature image.

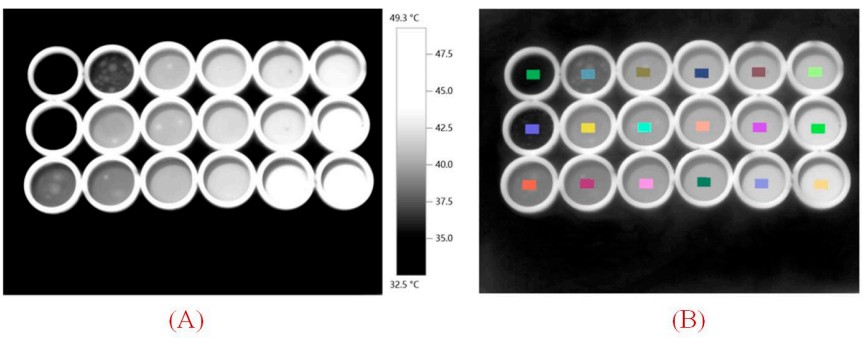

**Figure 4.** (**A**) Thermal infrared image on 3 August 2022 at 10:00; (**B**) Experimental sample (Colored rectangular squares: Selected experimental samples).

*2.2. OFT Inversion and Accuracy Evaluation*

In this section, the experiment was to establish the mapping relationship between the one-dimensional BTD data and the OFT data, so four classic inversion models were

constructed to invert OFT using the BTD; the training and testing sets of the model were the same. Additionally, three precision evaluation indicators were constructed to evaluate and analyze the inversion results.

### 2.2.1. Inversion Model Construction

(1) Regression Fitting Model

The relationship between OFT and BTD was established by taking the BTD as the independent variable and the OFT as the dependent variable. The BTD used in the regression was the average of the BTDs within each circle in the data set. Trying various regression models revealed that the exponential model was the best fit:

$$y = Ae^{\frac{x}{b}} + c \tag{1}$$

where $x$ is the BTD (°C); $y$ is the OFT (mm); and $A$, $b$, $c$ is the regression coefficient.

The data measured at 10:00 the next day of the summer and autumn experiments were regressed and fitted to obtain the curve shown in Figure 5. There was an exponential relationship between the BTD between oil and water and the thickness of the oil film. This conclusion was consistent with the conclusion reached in [33], but the fitting coefficient was different; this should be attributed to differences in experimental environmental conditions, time, and oil quality.

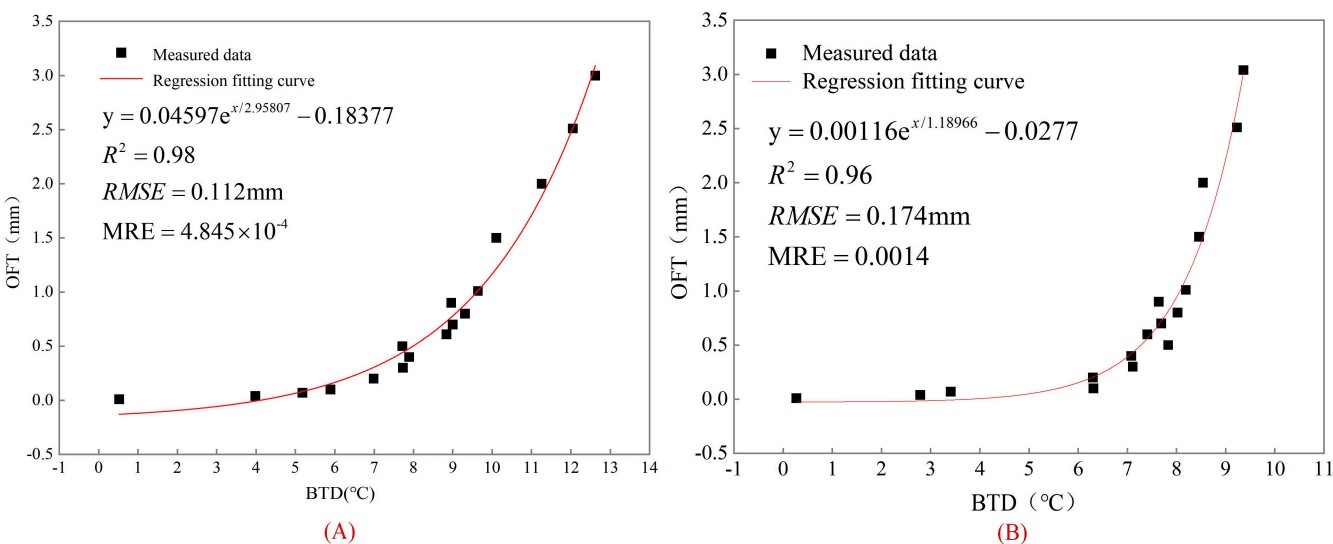

**Figure 5.** (**A**) Fitting curve of summer experiment; (**B**) Fitting curve of autumn experiment.

Regression analysis is performed at any point in time, and the regression coefficients needs to be determined for each time point:

$$y = A(t)e^{\frac{x}{b(t)}} + c(t) \tag{2}$$

where $A(t)$, $b(t)$, $c(t)$ is the regression coefficients at moment t of the day, varying with time.

(2) Random Forest (RF) model

RF is an algorithm that integrates multiple trees through the idea of ensemble learning, which is based on the decision tree, and the result with the largest number of votes is used as the output by voting on the results of the decision tree [36]. In the OFT inversion, 400 sample points were randomly selected for each thickness to participate in training, and the remaining sample points were involved in testing. The BTD corresponding to each sample point was used as input and the measured OFT value was used as output, and by constantly adjusting the model parameters, it was found that the model effect was best when the number of leaf nodes was 1 and the number of optimal trees was 500.

(3) Support Vector Regression model (SVR)

SVR is used to solve regression problems with Support Vector Machine (SVM). Vapnik et al. introduced an insensitive loss function on the basis of the SVM to obtain SVR, and its basic idea was to find an optimal classification surface so that all training samples had the smallest error from the optimal classification surface [37]. The input sample x is mapped to a high-dimensional space by a nonlinear map, and a linear model is built in this feature space to estimate the regression function [37]:

$$f(x, w) = w \cdot \phi(x) + b \tag{3}$$

where $w$ is the weight vector, $b$ is the threshold value. Using the insensitive loss function, the constraint optimization problem can be expressed as [37,38]:

$$min\frac{1}{2} \parallel w \parallel^2 + C\sum_{i=1}^{l} \left(\xi_i + \xi_i^*\right)$$
$$\text{s.t} \begin{cases} y_i - w \cdot \phi(x_i) - b \leq \varepsilon + \xi_i \\ -y_i + w \cdot \phi(x_i) + b \leq \varepsilon + \xi_i^*, i = 1, 2, \ldots, l \\ \xi_i \geq 0, \xi_i^* \geq 0 \end{cases} \tag{4}$$

where $\|w\|$ is the magnitude of the normal vector to the surface that is being approximated. $\xi_i, \xi_i^*$ are the slack variables. Through multiple trainings, it was found that the kernel function of SVR was RBF, and the value of the insensitive loss function was set to 0.01, the gamma function setting and penalty factor c were obtained by using the libsvm function to automatically optimize the parameters in the MATLAB environment, so that the effect of the model was optimized.

(4) Convolutional neural network model (CNN)

Convolutional Neural Network (CNN) is a typical deep learning algorithm. The experiment was to establish a mapping relationship between one-dimensional BTD with OFT; so, we chose 1D-CNN model, which consisted of an input layer, hidden layer, and output layer, of which the hidden layer included a convolutional layer, RELU layer, and fully connected layer. The data of the input layer was the normalized one-dimensional BTD data and the output data was the normalized OFT data. The model constructed in this paper included one input layer, two convolutional layers, two fully connected layers, and one output layer, where the output layer was set to regressionLayer.

The accuracy of the CNN model was related to the selection of parameters, which was determined by continuous experiments in the case of fixed training and test sets. The parameters to be determined included the number of convolution kernels of the first convolutional layer, the size of the convolution kernel of the first convolutional layer, the number of convolution kernels of the second convolutional layer, the size of the convolution kernel of the second convolutional layer, the number of neurons of the two fully connected layers, the number of batch training, the number of iterations, and the learning rate.

On the summer dataset, the model parameters were set to the number of convolution kernels of the first convolutional layer was twenty, the size of the convolution kernel was twelve, the number of convolution kernels of the second convolutional layer was thirteen, the size of the convolution kernel was one, the number of neurons of the first fully connected layer was forty, the number of neurons of the second fully connected layer was seventeen, the number of batch training was sixteen, the number of iterations was thirty-nine, and the learning rate was 0.008. On the autumn dataset, the parameters were set to fifteen, one, six, one, thirty-eight, eighteen, forty-eight, forty-three, and 0.003, respectively.

2.2.2. Model Accuracy Evaluation Index

In order to compare the OFT inversion accuracy of the four models, and explore the suitable OFT detection interval and the optimal OFT detection time of the day from the

inversion results, the accuracy evaluation indexes used in this paper include root mean square error (RMSE), mean relative error (MRE), and coefficient of determination ($R^2$).

$$RMSE = \sqrt{\frac{\sum_{i=1}^{n}(y_i - \hat{y}_i)^2}{n}} \tag{5}$$

$$MRE = \frac{1}{n}\sum_{i=1}^{n}\frac{|y_i - \hat{y}_i|}{\hat{y}_i} \tag{6}$$

$$R^2 = 1 - \frac{\sum_{i=1}^{n}(y_i - \hat{y}_i)^2}{\sum_{i=1}^{n}\left(y_i - \bar{y}\right)^2} \tag{7}$$

where, $y_i$ is the inverse OFT value (mm); $\hat{y}_i$ is the measured OFT value (mm); $\bar{y}$ is the average value of the inverted OFT (mm); and n is the number of test samples. *RMSE* and *MRE* reflect the degree of difference between the predicted value and the measured value, and the smaller the *RMSE* and *MRE*, the higher the accuracy; $R^2$ reflects the fitting degree of the model: the larger the value, the better the fitting effect.

## 3. Results and Analysis

This section first analyzes the observation results of ground experiments and analyzes the correlation between the OFT and the BTD. Then, based on the data of the moment with the greatest correlation, the inversion results of the four models are compared and analyzed to determine the suitable OFT inversion model. Based on this model, the inversion results of each moment are compared and analyzed, and the most suitable time period for detecting the OFT is determined.

### 3.1. The Relationship between BTD and OFT

3.1.1. Variation of BTD of Oil Film with Different Thickness in a Day

The BT changes in the water surface with and without oil film cover were divided into two groups to show the observations more clearly; there were four groups in autumn and summer (Figure 6). BTDs were, likewise, divided into four groups (Figure 7).

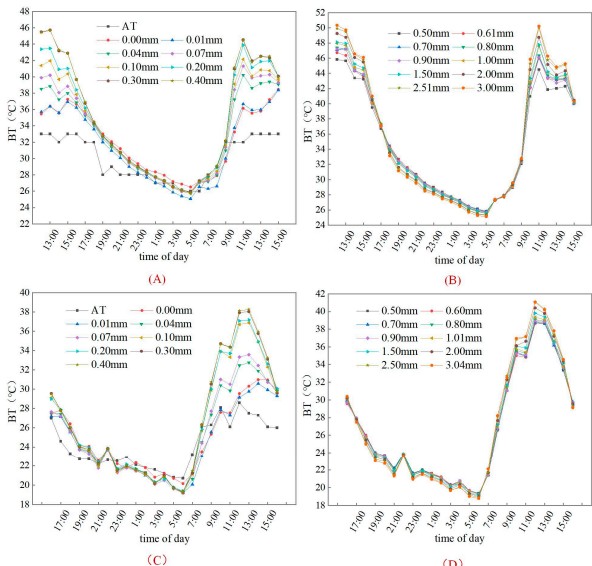

**Figure 6.** (**A**) Change of BT and air temperature (AT) of OFT (0.01–0.40 mm) in summer; (**B**) Change of BT of OFT (0.50–3.00 mm) in summer; (**C**) Change of BT and AT of OFT (0.01–0.40 mm) in autumn; (**D**) Change of BT of OFT (0.50–3.00 mm) in autumn.

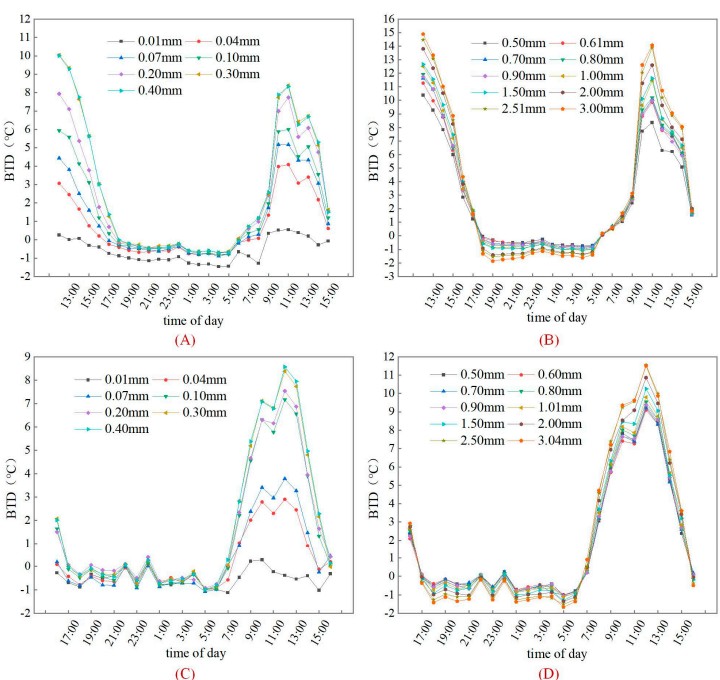

**Figure 7.** (**A**) Changes in brightness temperature difference of OFT (0.01−0.40 mm) in summer; (**B**) Change of BTD of OFT (0.50−3.00 mm) in summer; (**C**) Change of BTD of OFT (0.01−0.40 mm) in autumn; (**D**) Change of BTD of OFT (0.50−3.00 mm) in autumn.

As the temperature rose, the BT value on the surface of the oil film also increased, reaching a maximum at noon; the temperature decreased and the BT value decreased, except for small fluctuations in the middle. During the day, it conformed to the law that the thicker the oil film surface, the greater the BT value and the greater the BTD between oil and water, and this law was most obvious at noon. This is mainly because the thicker the oil film absorbed more solar heat, and the oil film covered the surface of the water, reducing the heat exchange between the water and the atmosphere and reducing heat loss. This is consistent with the conclusion in [33].

At night, due to the lack of sunlight, the BT value of the water surface covered by oil film decreased, it was found that there was not a simple monotonic relationship between OFT and BTD. As the OFT increased, the OFT was less than 0.30–0.40 mm and the BTD between oil and water became smaller; on the contrary, the OFT was greater than 0.30–0.40 mm, the conclusion was the opposite. This is mainly due to the smaller amount of oil per unit area of the thinner oil film. The main factors affecting the temperature variation of thinner oil films are evaporative cooling and solar heating. For thinner oil films, the thinner the oil film, the faster the evaporative cooling; on the contrary, the thicker the oil film, the slower the evaporative cooling. For thicker oil film, the thermal properties of the oil film play a major role: the BT value changed with the change of light; the greater the OFT, the more serious the heat lose. This indicates that 0.3–0.4 mm may be the dividing line of a thin and thick oil film.

### 3.1.2. Correlation Analysis between OFT and BTD

The optimal time of day to detect oil film depends on the correlation between OFT and BTD, and the greater the correlation, the better the detection of OFT. The correlation coefficient was used as a measure of the degree of linear correlation between the values of the study variable to analyze the relationship between OFT and BTD; the correlation coefficient between OFT and BTD was calculated for 24 moments in summer and 24 moments in autumn.

As shown in Figure 8, it could be seen from the summer results that during the day, the OFT and BTD showed a positive correlation. As the light intensity increased,

the correlation coefficient value increased. The correlation coefficient between OFT and BTD was the largest at 10:00 the next day, which was consistent with the conclusion obtained [33]. From 18:00 of the day to 5:00 of the next day (lack of sunlight), the OFT and the BTD showed a negative correlation, with the largest negative correlation value at 20:00 of the day. The conclusions reached in the autumn and summer were basically the same. From the correlation calculation results, it could be concluded that there was a strong correlation between OFT and BTD, so it was feasible to use BTD for OFT inversion.

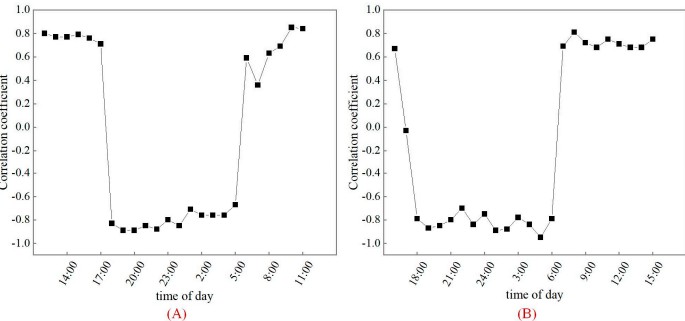

**Figure 8.** Correlation coefficient between OFT and BTD: (**A**) in summer; (**B**) in autumn.

### 3.2. Analysis of OFT Inversion Results

3.2.1. The Most Suitable Inversion Model

The data of the maximum correlation moment calculated in Section 3.1.2 (10:00 the next day in summer and autumn) was selected, the inversion results were divided into three OFT intervals and all OFTs for comparative analysis, and the model inversion accuracy and model stability were evaluated. The inversion accuracy of the model was the average of the results of ten runs.

On the summer and autumn datasets, the inversion accuracy of the four models were compared (Table 2). It could be seen from the inversion results of the entire OFT, and the CNN model performed best. In summer, the RMSE of the CNN model was 0.093 mm, which was 0.019 mm better than the regression fitting model, 0.018 mm higher than the RF model, and 0.016 mm higher than the SVR model. In autumn, the RMSE was 0.144 mm, which was 0.03 mm higher than the regression fitting model, 0.062 mm higher than the RF model, and 0.004 mm higher than the SVR model. Although it was found that the MRE of the RF model was smaller than that of the CNN model, the fitting effect of the CNN model was better than that of the RF model. For the inversion results of the three thickness intervals, the CNN model, the SVR model, and the RF model showed better inversion accuracy than the regression fitting model. Comparing the results of 10 runs of the three models (Figure 9), it was concluded that the CNN model and SVR model had strong model stability and high model operation efficiency while maintaining high inversion accuracy. Considering the oil film of the whole and individual thickness ranges, the CNN model was regarded as the best inversion model.

**Table 2.** Model inversion accuracy evaluation.

| OFT / Model | Accuracy Evaluation Indicators | Summer | | | | Autumn | | | |
|---|---|---|---|---|---|---|---|---|---|
| | | Regression Fitting | RF | SVR | CNN | Regression Fitting | RF | SVR | CNN |
| 0.01–0.07 mm | RMSE/mm | 0.085 | 0.011 | 0.019 | 0.010 | 0.059 | 0.006 | 0.013 | 0.006 |
| | MRE | 5.079 | 0.063 | 0.462 | 0.251 | 2.040 | 0.019 | 0.506 | 0.204 |
| | $R^2$ | −11.037 | 0.792 | 0.430 | 0.866 | −4.768 | 0.941 | 0.681 | 0.896 |
| 0.10–1.00 mm | RMSE/mm | 0.095 | 0.136 | 0.104 | 0.094 | 0.143 | 0.175 | 0.125 | 0.113 |
| | MRE | 0.241 | 0.169 | 0.174 | 0.153 | 0.279 | 0.265 | 0.239 | 0.201 |
| | $R^2$ | 0.893 | 0.749 | 0.854 | 0.860 | 0.755 | 0.615 | 0.801 | 0.818 |

**Table 2.** *Cont.*

| OFT / Model | Accuracy Evaluation Indicators | Summer | | | | Autumn | | | |
|---|---|---|---|---|---|---|---|---|---|
| | | Regression Fitting | RF | SVR | CNN | Regression Fitting | RF | SVR | CNN |
| 1.00–3.00 mm | RMSE/mm | 0.142 | 0.083 | 0.132 | 0.105 | 0.249 | 0.290 | 0.208 | 0.206 |
| | MRE | 0.056 | 0.011 | 0.047 | 0.042 | 0.100 | 0.091 | 0.088 | 0.095 |
| | $R^2$ | 0.959 | 0.983 | 0.955 | 0.973 | 0.879 | 0.823 | 0.909 | 0.906 |
| 0.01–3.00 mm | RMSE/mm | 0.112 | 0.111 | 0.109 | 0.093 | 0.174 | 0.206 | 0.148 | 0.144 |
| | MRE | 1.054 | 0.105 | 0.169 | 0.150 | 0.548 | 0.182 | 0.243 | 0.246 |
| | $R^2$ | 0.983 | 0.985 | 0.985 | 0.989 | 0.960 | 0.946 | 0.972 | 0.972 |

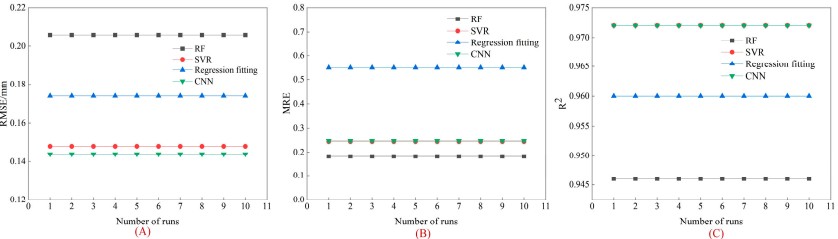

**Figure 9.** Model operation efficiency comparison based on autumn dataset: (**A**) RMSE; (**B**) MRE; (**C**) $R^2$.

### 3.2.2. Optimal Detection Time of Day for OFT

According to the conclusion of the previous section, the inversion results of the CNN model and SVR model at each moment were selected for comparative analysis. The inversion results of CNN and SVR are shown in Figures 10 and 11, respectively. From the summer and autumn inversion results, it could be concluded that the inversion results had high accuracy from 10:00 to 13:00, and the accuracy of the inversion results was the highest at 10:00 am the next day. In summer, the inversion results of the CNN model and the SVR model were, respectively, with RMSE values of 0.093 mm and 0.109 mm, $R^2$ values of 0.989 and 0.985, and MRE values of 0.150 and 0.169. In autumn, the inversion results had RMSE values of 0.144 mm and 0.148, $R^2$ values of 0.972 and 0.972, and MRE values of 0.246 and 0.243. In summary, whether in summer or autumn, around 10:00–13:00 during the day, when the oil film absorbs sunlight and begins to heat up and reach the maximum value of the day (the thermal balance between oil and water), is the optimal time to detect OFT. For nighttime data, the SVR model performed better than CNN model.

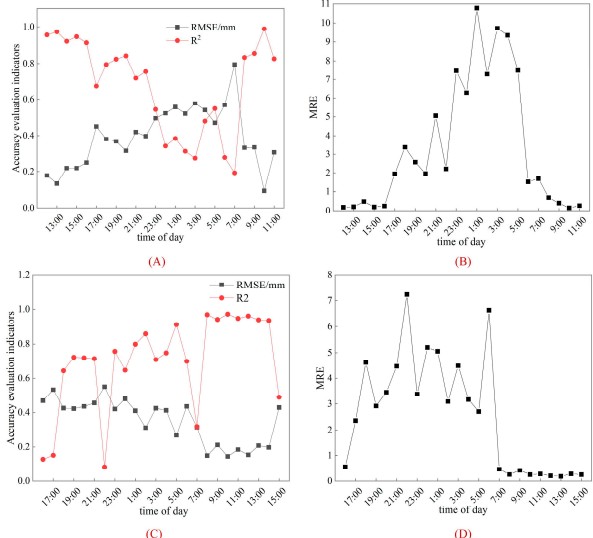

**Figure 10.** Comparison of accuracy of CNN inversion of OFT at different times: (**A**) RMSE and $R^2$ in summer; (**B**) MRE in summer; (**C**) RMSE and $R^2$ in autumn; (**D**) MRE in autumn.

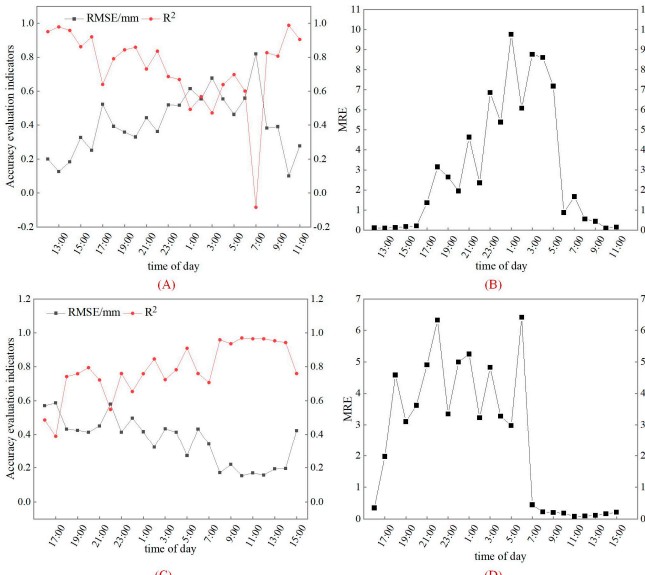

**Figure 11.** Comparison of accuracy of SVR inversion of OFT at different times: (**A**) RMSE and $R^2$ in summer; (**B**) MRE in summer; (**C**) RMSE and $R^2$ in autumn; (**D**) MRE in autumn.

## 4. Discussion

### 4.1. Detectable OFT Range Using Thermal Infrared Data

To study the range of OFT detectable by thermal infrared, the data at 10:00 the next day in summer and autumn was selected, and the CNN model was used to invert the OFT. The measured OFT value was compared and analyzed with the inverted OFT value. In this paper, the inversion results were visually represented by plotting scatter plots. The 1:1 scale line indicated that the measured value was equal to the inversion value, the inversion value above the 1:1 scale line meant that the inversion value was greater than the measured value, and the lower value meant that the inversion value was less than the measured value.

The results of the summer experiment were plotted in Figure 12; 0.1 mm error lines and 0.03 mm error lines were also plotted in Figure 12A. It could be intuitively seen that the OFT was greater than 1.00 mm, and the inversion result was within the error lines of 0.3 mm. It showed that the thermal infrared data was beneficial to invert the OFT greater than 1.00 mm. To show the experimental results more clearly, the inversion results of oil films of 0.01–0.10 mm thickness were plotted in Figure 12B, and those of 0.10–1.00 mm thickness were plotted in Figure 12C. The 0.01 mm and the 0.03 mm error lines were plotted in Figure 12B; the results showed that the OFT was less than 0.10 mm, the inversion results were mainly located outside the 0.03 mm error lines, and the thermal infrared data was not conducive to inverting the OFT of less than 0.10 mm. The 0.1 mm error lines and 0.02 mm error lines were plotted in Figure 12C. The inversion results of oil films of 0.20–0.30 mm thickness were mainly located outside the 0.2 mm error lines, and the thermal infrared data was not conducive to inverting the OFT of less than 0.3 mm. When the OFT was greater than 0.4 mm, the inversion result was within the error lines of 0.2 mm; therefore, the thermal infrared data was conducive to inverting the OFT greater than 0.4 mm. It could be seen from the summer data that the thermal infrared data was conducive to inverting the OFT greater than 0.4 mm.

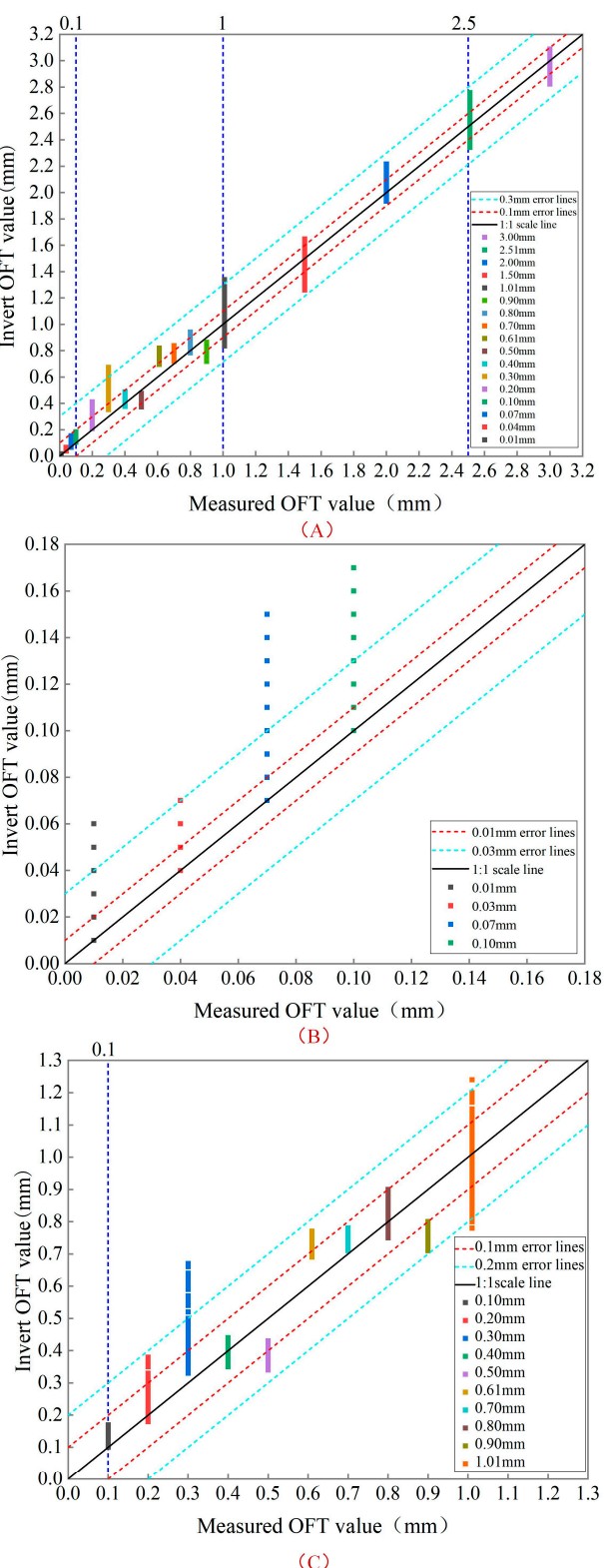

**Figure 12.** Comparison of measured OFT and invert OFT under summer dataset: (**A**) 0.01–3.00 mm; (**B**) 0.01–0.10 mm; (**C**) 0.10–1.01 mm.

The results of the autumn experiment are basically consistent with the conclusions of the summer experiment (Figure 13). In summary, thermal infrared is beneficial for detecting oil films with a thickness greater than 0.4 mm.

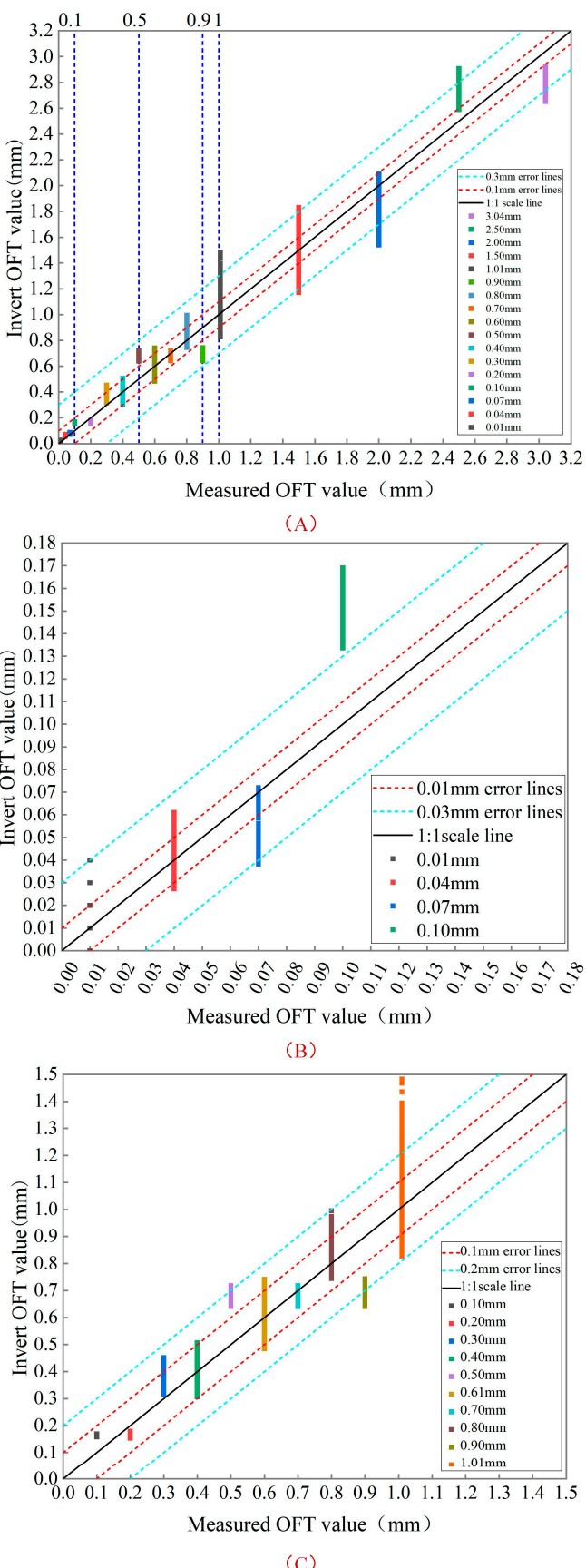

**Figure 13.** Comparison of measured OFT and invert OFT under autumn dataset: (**A**) 0.01–3.00 mm; (**B**) 0.01–0.10 mm; (**C**) 0.10–1.01 mm.

The OFT greater than or equal to 0.4 mm was selected for inversion of the data at 10:00 the next day in summer and autumn, and the CNN model was selected for the model. The inversion accuracy was shown in Table 3. For OFT greater than or equal to 0.40 mm, the MRE was 0.090 mm on the summer dataset; combined with Figure 14A, it could be seen that the inversion results were mainly located within the 0.1 mm error lines and some were located within the 0.3 mm error lines. On the autumn dataset, MRE was 0.117 mm; combined with Figure 14B, it could be seen that the inversion result was mainly located within the 0.3 mm error lines. Based on the above experimental results, it was concluded that thermal infrared was beneficial to invert the OFT greater than 0.4 mm, and the OFT greater than 0.4 mm was defined as a thicker oil film.

**Table 3.** Evaluation of inversion accuracy.

| Season | RMSE/mm | MRE | $R^2$ |
| --- | --- | --- | --- |
| summer | 0.099 | 0.090 | 0.986 |
| autumn | 0.162 | 0.117 | 0.962 |

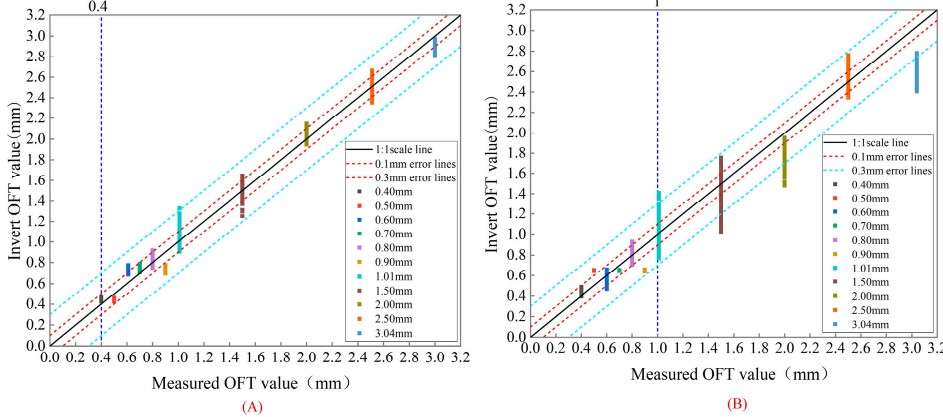

**Figure 14.** Comparison of measured OFT and inverted OFT: (**A**) summer, OFT $\geq$ 0.4 mm; (**B**) autumn, OFT $\geq$ 0.4 mm.

The advantage of thermal infrared over other optical sensors is its ability to detect the OFT at night. The data with the highest accuracy of the summer nighttime inversion (20:00 on 2 August) were selected; based on this data, the SVR model was used for inversion. Then, the analysis of the ability of thermal infrared to detect the OFT at night was carried out. The OFT was divided into three parts for inversion, the first part was to use the entire OFT for inversion, the second part was to use the thickness of 0.01–0.30 mm for inversion, and the third part was to use the thickness of 0.40–3.00 mm for inversion. As shown in Table 4, from the inversion results, it can be concluded that thermal infrared has a good detection ability for OFT greater than 0.4 mm.

**Table 4.** Evaluation of inversion accuracy (2 August, 20:00).

| OFT Range | RMSE/mm | MRE | $R^2$ |
| --- | --- | --- | --- |
| 0.01–3.00 mm | 0.330 | 0.169 | 0.859 |
| 0.01–0.30 mm | 0.069 | 0.557 | 0.535 |
| 0.40–3.00 mm | 0.161 | 0.105 | 0.963 |

The inversion results were compared with the measured OFT value, the 1:1 scale lines and error lines were also drawn, and then each OFT was analyzed separately. It could be seen from Figure 15A that for OFT of 0.01–0.30 mm, the inversion results were mainly located outside the error lines of 0.01 mm and 0.05 mm, and thermal infrared had poor detection ability for oil films in this thickness range. As shown in Figure 15B, for OFT

greater than 0.4 mm, most of the inverted OFT values were greater than the measured OFT values, most of which were within the error line; some would also be located outside the error line. In summary, at night, thermal infrared can detect OFT greater than 0.4 mm, but the detection effect will be insufficient compared with daytime.

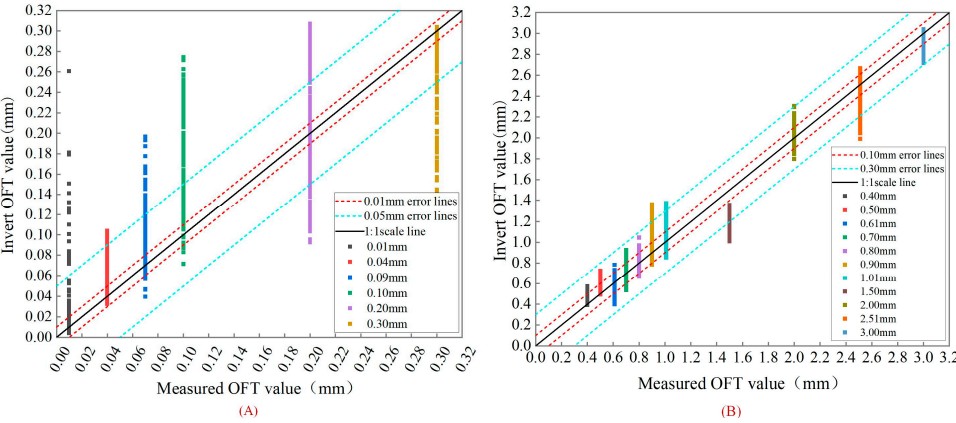

(A)　　　　　　　　　　　　　　　　　　　　(B)

**Figure 15.** Comparison of measured OFT and inverted OFT at 20:00 on 2 August: (**A**) 0.01–0.30 mm; (**B**) 0.40–3.00 mm.

### 4.2. Oil Spill Amount Estimation under Simulated Oil Spill Scenarios

The setting of the OFT experimental scene proved the ability of thermal infrared to detect the OFT, and the oil spill scene was added to verify the applicability of the experimental results. In the experiment, 5 g of crude oil was dripped into the water tank, data from two moments in the same oil spill scenario were used for experiments, and the SVR model was used to invert the OFT. We estimated the amount of oil spill and plotted the OFT grade. The data at the first moment was inverted, and the oil spill was estimated to be 3.423 g (MRE = 0.315); the OFT distribution was shown in Figure 16C. The oil film was distributed as a thick middle edge and a thin edge, which was in line with the actual process of oil spill diffusion. The data at the second moment was inverted, and the amount of oil spill was estimated to be 5.534 g (MRE = 0.107); the OFT distribution was shown in Figure 17C. The boundary of the oil film distribution was not obvious enough, because the OFT at this moment was relatively thin; thermal infrared was not sensitive enough to thinner oil films.

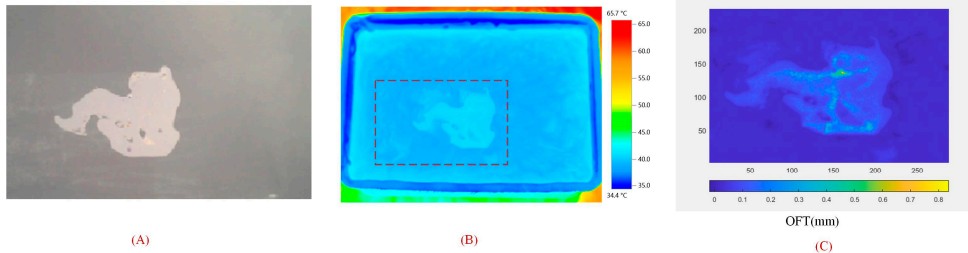

(A)　　　　　　　　　　　　　(B)　　　　　　　　　　　　　(C)

**Figure 16.** (**A**) RGB image of oil spill scene at 15:27 on 3 August 2022; (**B**) thermal infrared image of oil spill scene at 15:27 on 3 August 2022; (**C**) OFT distribution map at 15:27 on 3 August 2022.

In the actual oil spill scenario, due to the changeable maritime environment, there was still a certain gap in the application of the conclusions under laboratory conditions in reality. It was necessary to establish a large number of thermal infrared data sets of OFT in different environments for the estimation of oil spills, and it was beneficial to estimate the oil spill in different scenarios by establishing a physical model to simulate the BT value combined with environmental conditions.

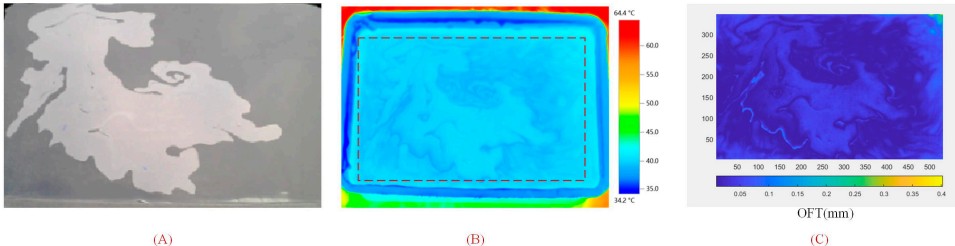

**Figure 17.** (**A**) RGB image of oil spill scene at 15:28 on August 3, 2022; (**B**) thermal infrared image of oil spill scene at 15:28 on 3 August 2022; (**C**) OFT distribution map at 15:28 on 3 August 2022.

## 5. Conclusions

OFT estimation is of great significance for the emergency treatment and risk assessment of oil spill accidents, and thermal infrared has advantages in OFT detection. Through the design of ground experiments, the application of thermal infrared in OFT detection was explored, and the quantitative inversion of OFT was realized by establishing the relationship between BTD between oil and water and OFT; four inversion methods (Regression fitting model, SVR model, RF model, and CNN model) were included.

In response to the four questions raised earlier, this paper draws four main conclusions: (1) In order to verify the inversion accuracy of four models, the data corresponding to the time with the greatest correlation between OFT and BTD was selected for inversion; the experimental results show that the CNN model has the advantages of high stability while maintaining high-precision inversion, and can be used as the preferred model for OFT inversion. (2) The BT value of the OFT in the day was constantly changing, and the optimal OFT detection time is about 10:00 to 13:00 in the day, which is not affected by seasonal changes. (3) In order to explore the OFT ranges that could be inverted by thermal infrared, the data corresponding to the optimal detection time of OFT was chosen, and the inversion results obtained by the CNN model were compared with the measured OFT value. The results show that thermal infrared has good detection ability for OFT greater than 0.4 mm during the day, but weak detection ability for thinner oil films. (4) The OFT detection capability of thermal infrared at night was also studied. At night, thermal infrared has a weak detection ability for thinner oil films and has a certain detection ability for relatively thick oil films, but the accuracy is lower than that in the daytime.

This experiment was carried out in a relatively controlled environment with sunny summer and autumn weather. The maritime environment is complex and changeable, and there are still great uncertainties in applying the experimental conclusions to the real marine environment. In order to use this constructed model for field experiments, it is also necessary to obtain thermal infrared OFT datasets in different scenarios for further examination. In the future, the team will carry out experiments under different environmental conditions; more complex scenarios are built to achieve more effective oil film thickness estimation.

**Author Contributions:** Conceptualization, M.W., J.Y. and S.L.; methodology, M.W., S.L. and J.Y.; software, M.W., S.L. and J.Y.; validation, M.W. and J.Y.; formal analysis, M.W.; data curation, M.W. and J.Y.; writing—original draft preparation, M.W.; writing—review and editing, M.W., J.Y. and S.L.; supervision, J.Z., Y.M. and J.W.; project administration, J.Z., Y.M., S.L. and J.Y.; funding acquisition, J.Z., Y.M. and J.Y. All authors have read and agreed to the published version of the manuscript.

**Funding:** This research was funded by the National Natural Science Foundation of China (No. U1906217, No. 61890964, No. 42206177, No. 42076182), Shandong Provincial Natural Science Foundation (No. ZR2022QD075), Fund of Technology Innovation Center for Ocean Telemetry, Ministry of Natural Resources (No. 2022004), Qingdao Postdoctoral Application Research Project (No. qdyy20210082), the Fundamental Research Funds for the Central Universities (No. 21CX06057A).

**Data Availability Statement:** Data underlying the results presented in this paper are not publicly available at this time but may be obtained from the authors upon reasonable request.

**Acknowledgments:** The authors thank the reviewers and editors for their positive and constructive comments, which have significantly improved the work.

**Conflicts of Interest:** The authors declare no conflict of interest. The funders had no role in the design of the study; in the collection, analyses, or interpretation of data; in the writing of the manuscript, or in the decision to publish the results.

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
