# Peer review of "Quantitative Inversion Ability Analysis of Oil Film Thickness Using Bright Temperature Difference Based on Thermal Infrared Remote Sensing: A Ground-Based Simulation Experiment of Marine Oil Spill"

_remotesensing, doi:10.3390/rs15082018_

Round 1

Reviewer 1 Report

This paper proposes quantitative inversion of marine oil film thickness using bright temperature difference based on thermal infrared remote sensing. While the authors have illustrated their method in detail and demonstrated using bright temperature difference based on thermal infrared remote sensing images is useful for quantitative inversion, I still have some concerns that need to be addressed.

1. The abstract part needs to further highlight the authors’ own work, and the collected data can also be introduced as part of contribution.

2. In the introduction part, some advantages and disadvantages of using infrared inversion need to be added. For the disadvantages, whether the method in this article is involved, and if so, give a brief description.

3. Need to increase the used analysis methods, the three mentioned in the article are too single and simple.

Reviewer 2 Report

The authors present a technique for estimating the oil film thickness on the water by using thermal sensors.  The paper is interesting and fit for publishing after correcting a few minor details.

The text labels on figures 5 onward are too small, which makes them hard to read.  Please increase the font size and increase the resolution and size of the figures, it's hard to track each line due to the small size and blur.

l. 258 "had strong model stability": How strong?  From figure 9 it seems that nothing changed from one run to the next.  If there is any difference, please quantify.  Otherwise, please explain why don't the error metrics change from one run to the next.

Reviewer 3 Report

 In this paper, the outdoor oil spill simulation experiments were designed,some models were applied to the inversion of OFT. That is a good job. Some suggestion is as follows:

1. line 83, "wind conditions less than level 4", different wind level should be added.

2. Information of Anemometer,such as Brand, precision et al.

3. lines 155-156 misses reference mark

4. line 164 Formula parameters need to be detailed

Reviewer 4 Report

Review to manuscript entitled “Quantitative inversion of marine oil film thickness using bright temperature difference based on thermal infrared remote sensing: A ground-based simulation experiment of marine oil spill” by Meiqi Wang, Junfang Yang and colleagues to the Remote Sensing. The authors designed an outdoor oil spill simulation experiment where the bright temperature data of different Oil film thickness in summer and autumn.

The subject of the manuscript seems to me of good relevance, but I do not believe that this manuscript can be published as it is and therefore, I recommend major revisions. It is critical that they review the points listed below:

1)    In Figure 4(b) - What is the meaning of the colored squares in the center of each sample? There is nothing about these colored squares in the caption or in the text when describing the figure.

2)    As explained in the text, different masses of crude oil were dripped into rings to form different OFTs and then, to minimize the effect of shadows caused by the rings, a smaller area was selected to perform the analysis. The problem is that depending on the material of the rings, the effects on the temperature will not only be because of the shadows but in the heat storage capacity of the ring itself and this would affect the results. How to ensure that these rings do not become a source of heat for the oil?

3)    As stated in line 200 - This is mainly because the thicker the oil film surface oil film absorbed more solar heat, and the oil film cover the surface of the water, reducing the heat exchange between the water and the atmosphere and reducing heat loss Could the justification for this be the difference in volumetric heat capacity between the oil and the water rather than this reduced heat exchange between the water and the atmosphere?

4)    In Figure 6 - I believe that the water temperature of the tank should be in the data. Don’t the authors think that the influence of the tank water temperature is not relevant? The rings where the oil is stored are in a thermal bath with the water, correct?

5)    It is not clear what the purpose of using Random Forest (RF) model and Support Vector Regression model (SVR). What did the inversion methods added to the results or to the study of the relationship between OFT and BTD?

 6)   Quantitative inversion was not the focus of this work, and they did not detail enough it to make it clear to the reader its importance to the results, I believe that they should change the title of the article.

Round 2

Reviewer 4 Report

The authors have performed the suggested adjustments and I believe the article is ready to be published in its current form.